# Recognition of Damage Modes and Hilbert–Huang Transform Analyses of 3D Braided Composites

**Gang Ding [1,2], Liankun Sun [3], Zhenkai Wan [1], Jialu Li [1,4], Xiaoyuan Pei [1,4,\*] and Youhong Tang [5,\*]** 

[1] School of Textiles, Tianjin Polytechnic University, Tianjin 300387, China; dinggang@tjpu.edu.cn (G.D.); wanzhenkai@tjpu.edu.cn (Z.W.); lijialu@tjpu.edu.cn (J.L.)

[2] Supervisory and Evaluation Office, Tianjin Radio & TV University, Tianjin 300191, China

[3] School of Computer Science and Software Engineering, Tianjin Polytechnic University, Tianjin 300387, China; sunliankun@tjpu.edu.cn

[4] Composites Research Institute, Tianjin Polytechnic University, Tianjin 300387, China

[5] Centre for Nano Scale Science and Technology, and School of Computer Science, Engineering, and Mathematics, Flinders University, Adelaide, SA 5042, Australia

\* Correspondence: pei_xiaoyuan@126.com or peixiaoyuan@tjpu.edu.cn (X.P.); youhong.tang@flinders.edu.au (Y.T.)

**Abstract:** The identification and classification of acoustic emission (AE) based failure modes are complex due to the fact that AE waves are generally released simultaneously from all AE-emitting damage sources. To fully understand the occurrence of damage and the damage evolution law of 3D braided composites, the tensile response characteristics and failure mechanisms of such composites were revealed by experiments, followed by frequency domain analyses. The results indicated good correlation between the number of AE events and the evolution of damage in 3D braided composites. After an AE signal was decomposed by the Hilbert–Huang transform (HHT) method, it might extract and separate all damage modes included in this AE signal. Additionally, the frequency saltation in the HHT spectra implied changes in the failure mode of the 3D braided composites. This study provides an effective new method for the analysis of the tensile fracture mechanism in 3D braided composites.

**Keywords:** polymer-matrix composites; numerical analysis; acoustic emission; Hilbert–Huang transform (HHT)

## 1. Introduction

High-performance polymer matrix composites have been widely used in the aviation, aerospace, automotive, and building engineering fields [1]. Over a long service period, these composites inevitably experience damage and failure [2–4]. Benefiting from its dynamic, sensitive, real-time, and anti-interference characteristics, acoustic emission (AE) technology has been gradually applied to the detection of damage in composite materials [5]. Researchers have conducted many studies of the use of AE technology in damage detection in composite materials. Dickinson and Fletcher described in detail the acoustic detection of invisible damage in aircraft composite panels [6]. Carvelli et al. used the AE features of polyphenylene sulphide carbon woven composites during loading and developed understanding of the damage mechanism [7]. Baccar and Soeffker examined the AE signals generated from laminated carbon fiber reinforced polymer (CFRP) subjected to an indentation test. The continuous wavelet transform was applied to AE signals in order to identify and classify failure modes in laminated composites [8]. During tensile tests on two ceramic matrix composites, a new waveform-based procedure was proposed for the selection of AE events generated by damage [9]. The procedure included accurate localization and selection assessment based on signal energy.

The yarns of 3D braided composites interweave in space and form a spatial network structure that can directly produce a variety of complex shapes to eliminate the process of cutting to form joints, overlaps, and splices [10–12]. Because of the anisotropy and heterogeneity of braided composites, it is very difficult to characterize identification of the evolution of damage [13]. Zhang et al. conducted a meso-scale finite element analysis (FEA) to predict the mechanical properties and simulate the progressive damage of 3D braided composites under external loadings [14–16]. Ivanov et al. studied failure analysis of a triaxial braided composite. Tensile tests were instrumented with optical surface strain and AE measurements. The results showed that the damage developed in two stages: (1) intra-yarn cracking that increased crack densities and crack lengths and (2) local inter-yarn delamination and conjunction of intra-yarn cracks [17]. Combining AE parameters and compression after impact, Yan et al. studied the failure processes and fracture mechanisms of 3D braided composites [18]. In the literature, several methods have been applied to extract damage features from AE signals and then carry out signal processing and pattern recognition [6–9,17,18]. The above methods have their respective advantages and disadvantages in recognition of the damage patterns of composite materials.

Due to the essential characteristics of AE signals and the superposition of non-homologous signals, the resulting damage signals are often accompanied by nonstationary and nonlinear characteristics. Huang et al. in 1998 proposed the Hilbert–Huang transform (HHT), a nonlinear and nonstationary signal analysis and processing method [19]. The HHT is composed of two main theoretical parts: empirical mode decomposition (EMD) and Hilbert spectral analysis. Depending on the characteristics of the signal, EMD adaptively decomposes any complex signal into a series of intrinsic modal functions (IMFs). The IMF has the following two features: the quantity of signal maxima is equal to the number of zeros or the difference is 1, and the local mean of the signal is zero, which is the upper envelope defined by the maximum and the lower envelope defined by the minimum. After EMD, the instantaneous frequency of each IMF can be calculated by the HHT. From the above two steps, the original signal can be expressed as a 3D time-frequency-energy distribution, called the Hilbert spectrum. On the basis of the Hilbert spectrum, the marginal spectrum can be obtained by means of integral. Starting from the local characteristics of signals, the HHT constructs the basic functions directly on the signal itself, thereby obtaining decomposition components at different scales. Especially after introducing the idea of resolution, the HHT overcomes the modal aliasing that can occur in the decomposition of AE signals [20]. The HHT has good visualization and local adaptability and can reveal the time-frequency characteristics of signals. It has been widely recognized in nondestructive testing (mainly in the detection of AE) signal processing. The HHT can carry out signal analysis and data processing for AE detection of signals in different materials, e.g., cement, metal, composite, and so on. Han and Zhou used EMD and HHT to study the frequency distribution of AE signals to gain deeper understanding of the initiation, growth, and evolution of different types of damage [21]. Han et al. also performed tensile testing of a CFRP laminate along with AE monitoring [22]. The HHT of AE signals can accurately calculate instantaneous frequencies for recognition of damage modes to help understand the damage process. Hamdi et al. established an AE pattern recognition approach based on the HHT on glass fiber reinforced polymer composites, but only the first intrinsic mode, IMF1, was chosen as the descriptor of failure modes, other IMF components being ignored in failure mode recognition [23]. Zhang et al. used the HHT to analyze the failure mechanism of 3D braided composites on low-velocity impact [24]. Their results showed that frequency saltation in the HHT spectra implied changes in the failure mode of the composites.

At present, AE technology has become an effective means of nondestructive testing and damage assessment of composite materials. However, the particular characteristics of 3D braided composites, including self-inhomogeneity, anisotropy, and complexity of manufacturing process, make the AE signals of 3D braided composites very complex. Therefore, the result of AE signal processing is directly related to the effectiveness of the AE testing. For a stationary signal, only the time domain characteristics or frequency domain characteristics of the signal are considered to reflect

the state of the signal. However, for a typical non-stationary signal of a three-dimensional braided composite acoustic emission signal, it is not only necessary to describe the change of the signal frequency with time, but also a time-frequency joint function is needed to represent the signal, i.e., the time-frequency representation of the signal. As an important method to deal with non-stationary random signals, the time-frequency analysis can effectively identify and remove the interference noise in the signal through the time-frequency spectrum, and comprehensively identify the characteristics of the signal. Typical time-frequency analysis methods include short-time Fourier transform (STFT), wavelet transform (WT), S-transform (ST), Gabor expansion, and Hilbert–Huang transform (HHT) and the like.

Compared with the other four time-frequency analysis methods, HHT has a good local adaptability and intuitiveness of analysis results. It has a greater progress in resolution than traditional time-frequency analysis methods, so it is more suitable for three-dimensional braided composite acoustic emission signals. Time-frequency analysis can reveal the time-frequency characteristics unique to acoustic emission signals. The HHT is based on AE signals and does not include any human disturbance. It is suitable for the analysis of nonlinear and nonstationary signals, having strong adaptability to the signal, and can well reflect the local frequency characteristics of the signal. It is very suitable for time-frequency analysis of the AE signals of 3D braided composites. To reveal the tensile damage model for 3D braided composites, the characteristics of AE signal time and frequency domains are used to build the corresponding relation between the damage modes and AE signals.

In this study, the tensile tests of three groups of 3D braided composite specimens with the same braiding angle but different fiber volume contents were carried out, along with AE monitoring. In the tests, the mechanical performance and the AE signal at different damage stages were obtained. In order to study the time-frequency characteristics of AE signals, the HHT time-frequency analysis method was used to separate and extract damage pattern information. Accordingly, the corresponding relation between damage modes and AE signals of 3D braided composites was obtained. An AE signal was decomposed by HHT, effectively separating and extracting all damage modes of the 3D braided composites included in this AE signal. This study provides a new signal analysis method for the study and analysis of the tensile fracture mechanisms of 3D braided composites and an effective method for safety monitoring of 3D braided composites using the AE technique.

## 2. Experimental Procedure

### 2.1. 3D Braided Composites

Preforms of 3D braided composites were prepared by a four-step $1 \times 1$ braiding procedure, using T700-12K carbon fiber for the 3D braided preform. The density of the T700 carbon fibers was 1.76 g/cm$^3$. Before curing, the preforms of the 3D braided composite were placed in a cool ventilated place and dried at room temperature to achieve a stress balance. The 3D braided composites were prepared by a resin transfer molding (RTM) method. The interlacement and curing process of the 3D braided composites are shown in Figure 1. Epoxy resin (TDE-85, Sigma, St. Louis, MO, USA) combined with a curing agent (*N*,*N*-Dimethyl benzyl amine, Sigma, St. Louis, MO, USA) and an accelerating agent (HK-021 Me-THPA, Sigma, St. Louis, MO, USA) was injected into the preform by the RTM method. The curing cycle was 130 °C for 2 h, 150 °C for 1 h, 160 °C for 8 h, and finally 180 °C for 3 h. Each specimen was about 250 mm long × 25 mm wide × 4 mm thick. The sensor locations for each specimen are shown in Figure 2. The process parameters and names are given in Table 1.

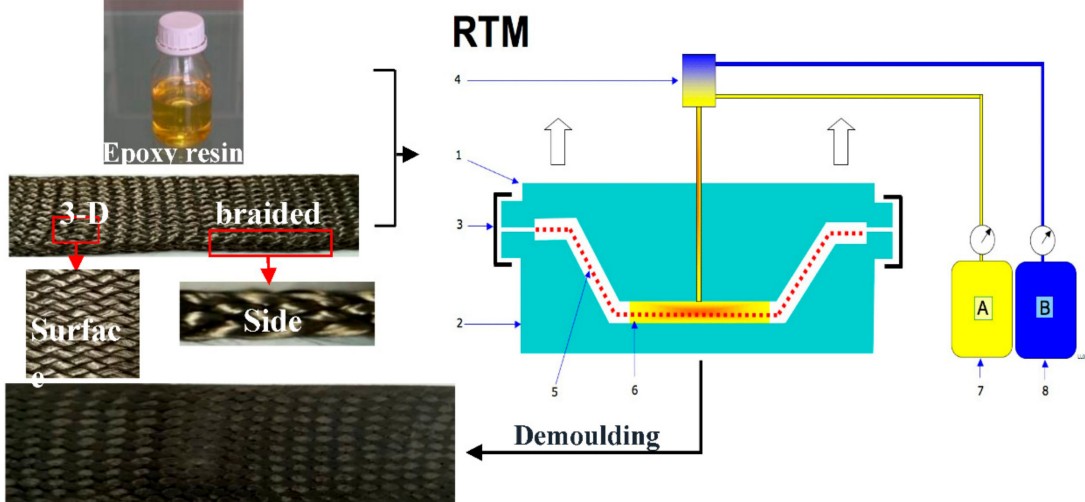

**Figure 1.** Design and resin transfer molding (RTM) curing process of a 3D braided composite [25]. 1 and 2: upper and lower molds, respectively; 3: mold sealing device; 4: storage tank and entry point of resin and curing agent; 5: non-infiltrated fabric; 6: fabric that has been impregnated with resin; 7 and 8: storage bins for resins and curing agents, respectively.

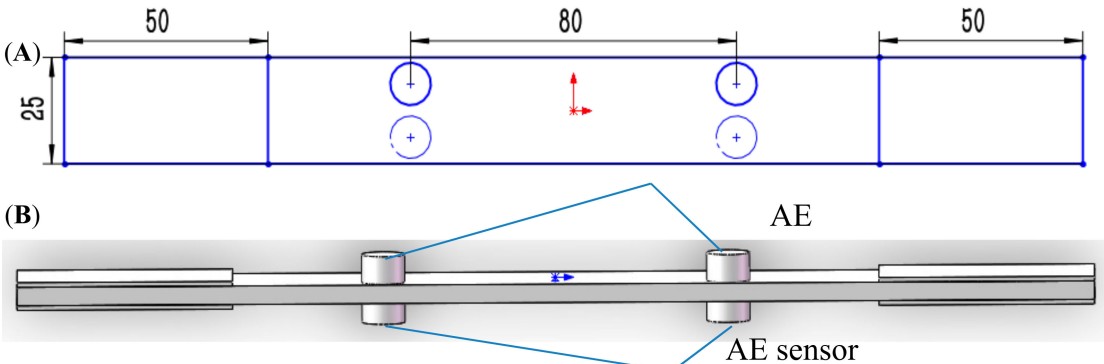

**Figure 2.** Geometry and sensor location for each specimen with (**A**) top and (**B**) side views. AE: acoustic emission.

**Table 1.** Dimensions of each specimen.

| Specimen | Surface Braiding Angle ($\alpha$)/° | Internal Braiding Angle ($\gamma$)/° | Pitch Length (h)/mm | Fiber Volume Fraction % |
|---|---|---|---|---|
| B-V1 | 22.60 | 32.14 | 6.0 | 45.00 |
| B-V2 | 21.60 | 30.69 | 5.0 | 52.50 |
| B-V3 | 22.60 | 32.14 | 3.5 | 60.91 |

## 2.2. AE Signal Acquisition System

Damage to 3D braided composites under tensile loading can be analyzed by AE signals. The AE data-acquisition system manufactured by Shenghua Corporation, China, was used to record AE activity. The AE sensors used were broadband sensors with an operating frequency range of 100–1000 kHz. Grease was used to achieve good acoustic coupling on the surface of the sensor. To avoid distortion of the sampled signals caused by spectral aliasing, the sampling frequency of the AE device was set at 900 kHz in accordance with the law of sampling. The highest filtering frequency of the low pass filter was 20 kHz and the lowest filtering frequency of the high pass filter was 1 MHz. The MV voltage signal was output by an AE sensor connected to the differential input of the preamplifier. The threshold value was set at 45 dB so that AE signals above this value were recorded. Test acquisition parameters included event count, amplitude, hit and hit count, duration, count, energy, rise time, root mean square (RMS) voltage, and arrival time.

Before tensile testing of the 3D braided composites, a breaking lead test was used to determine the speed of sound at the experiment. When the magnitude of the measurement was greater than 85 dB it was considered to meet the qualification. Otherwise, proper adjustment was required of external conditions, such as vacuum grease, until the standard was met. The tensile specimens were manufactured in accordance with ASTM standards [26]. To study the AE characteristics of 3D braided composites in tension tests, specimens with different braiding parameters were selected. The dimensions of each specimen are listed in Table 1. To avoid testing error, specimens with each parameter were tested 5 times. The tensile process of a 3D braided composite is shown in Figure 3.

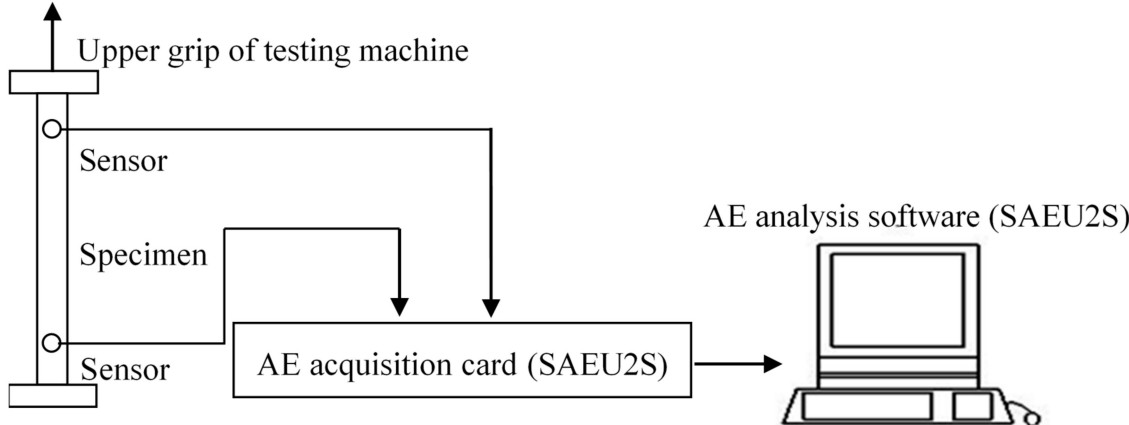

**Figure 3.** The tensile process of the 3D braided composites.

## 3. Hilbert–Huang Transform

The HHT is a nonlinear and nonstationary signal analysis and processing method that is completely independent of the Fourier transform. It relies on the characteristics of the signal itself to implement the transformation, eliminating the influence of human factors.

### 3.1. The Basic Principles of EMD

The core of the EMD process is continuous "screening". The detailed decomposition of any signal $X(t)$ is shown as follows:

(1) All local maxima and local minima of $X(t)$ are determined;
(2) All local maxima are synthesized on the envelope $X_{max}(t)$ and all local minima are synthesized against the lower envelope $X_{min}(t)$ by a cubic spline;
(3) The mean of the upper and lower envelopes is calculated by Equation (1):

$$m_1 = [X_{max}(t) + X_{min}(t)]/2. \tag{1}$$

(4) The mean $m_1$ is eliminated from the original data sequence $X(t)$ and a new data sequence $h_1(t)$ is obtained using Equation (2):

$$h_1(t) = h_1(\text{t}) - m_1(t). \tag{2}$$

(5) Determine whether $h_1(t)$ satisfies the IMF condition: If it is satisfied, $h_1(t)$ is the IMF. If it is not satisfied, loop execution steps (1)–(4) to obtain a suitable $h_1(t)$, until $h_{1k}(t)$ satisfies the IMF condition, the first order eigenmode function decomposed $C_1$ is obtained.

$$H_{11} = h_1 - m_{11}$$
$$\cdots \cdots$$
$$C_1 = h_{1k} = h_{1(k-1)} - m_{1k} \tag{3}$$

Thus, the process of extracting the first eigenmode function is complete.

Next, the component $C_1$ is separated from the original signal.

$$X(t) - C_1 = r_1 \tag{4}$$

$r_1$ is treated as the new data according to the above same steps, decomposing the n order IMF and a residual component $r_n$. The residual component $r_n$ is an average trend or a constant of the original data. It cannot be decomposed by EMD again.

Equation (4) is further calculated and Equation (5) can be obtained:

$$X(t) = \sum_{i=0}^{n} IMF_i(t) + r_n(t) \tag{5}$$

where $r_n(t)$ is the residual component that depicts the overall trend of the signal curve.

Here, it is obviously not convenient to decide when to stop filtering directly through the definition of IMF. So Huang defines standard deviation (*SD*) as the criterion for determining when the process will end.

$$SD = \sum_{t=0}^{T} \left[ \frac{|h_{n-1}(t) - h_n(t)^2|}{h_{n-1}^2(t)} \right] \tag{6}$$

If the standard deviation *SD* is in the range of 0.2–0.3 after two continuously screening, the screening process can be stopped.

## 3.2. HHT Analysis

After each IMF is the HHT, the analytic signal is shown in Equation (7):

$$X(t) = Re \sum_{j=0}^{n} a_j(t) e^{j\varphi_1(t)} = Re \sum_{i=0}^{n} a_j(t) e^{j \int \omega_1(t)dt} \tag{7}$$

where *Re* is the operation used to extract the real part. In Equation (7), $a_j(t)$ represents an analytic signal, *t* is time. The right side of Equation (7) is the Hilbert time-frequency spectrum, that is:

$$H(\omega, t) = Re \sum_{i=0}^{n} a_j(t) e^{j \int \omega_1(t)dt} \tag{8}$$

Using Equation (8), the amplitude is depicted in the form of contours in the time-frequency plane. For further calculations, the Hilbert marginal spectrum is obtained by Equation (9).

$$h(\omega) = \int_0^T H(\omega, t)dt \tag{9}$$

In Equation (9), $H(\omega, t)$ accurately describes the change rule of the amplitude of a signal over time and frequency across the frequency band.

Advantages of HHT

Compared with other time-frequency analysis methods, the HHT has many advantages. Traditional data processing methods, such as the short-time Fourier transform (STFT), wavelet transform, or S-transform, can only deal with linear and nonstationary signals. The Wigner–Ville distribution is a nonlinear analysis method in form, but a large number of cross terms are generated in the transformation process, affecting the analysis results. The HHT differs from these traditional methods in that it is completely free of the bondage of linearity and stability. It is suitable for analyzing nonlinear and nonstationary signals. It can generate IMFs from the screening process that differ from the STFT, wavelet transform, and S-transform. The HHT can also achieve very high precision in

both time and frequency, making it very suitable for analyzing abrupt signals. The phase function is first obtained by the HHT and then the instantaneous frequency is generated by derivation of the phase function.

## 4. Results and Discussion

### 4.1. Experimental Results and Discussion

The mechanical properties of 3D braided composites depend not only on the properties of fiber, matrix, and interface, but also on their internal structures. Tensile testing was carried out with AG-250KNE SHIMADZU universal material experiment machine. The experiment machine is controlled by computer and has high precision. Three kinds of fiber volume content composites were used in the experiment, as listed in Table 1. There were five specimens for composites with the same structural parameters. Table 2 lists the main mechanical parameters of 3D braided composites in tensile testing when the specimen is stretched axially. Typical tensile stress-strain curves are shown in Figure 4. It can be seen from Table 2 and Figure 4 that when the braiding angle of the braided composite is held constant, the tensile strength and tensile modulus of specimens increases with the increase in fiber volume fraction. Because the fiber bundle is the reinforcement, it bears the main load and the resin matrix acts to transfer the load. In the tensile process, a thin layer of resin on the surface of the specimen firstly produced the crack; then the cracks proliferate and proliferate; finally, the specimen breaks down in a short time. In addition, in the tensile test, when the load reached a certain value, the specimen started to make a slight noise. With the increase of the load, the specimen suddenly broke down. This shows that the fiber bundle is not gradually broken, but it is almost pulled apart at the same time. These are two modes of damage: one is the crack extended along the fiber bundle, the other is that the fiber bundle is broken. Additionally, the damage in 3D braided composites is mainly a fracture of the fiber bundles. When the braiding angle is held constant, the fiber bundle that withstands load increases with an increase in fiber volume, and the overall carrying capacity of the fiber bundle also increases. When the specimen with the same volume is stretched, the fiber volume fraction is larger, the more fiber bundles can bear the load. The tensile failure mode of 3D braided composites is mainly the fracture of fiber bundles. The tensile mechanical properties of the composites were improved by fiber volume fraction. It is also evident from Figure 4 that at the beginning of the tensile test, the tensile stress of the specimen increases rapidly in the tensile direction. After that, the stress gradually increases linearly until final fracture of the specimen. The fracture of the 3D braided composites is a brittle fracture. The nonlinear change at the beginning may be due to micro defects such as impurities in the specimen [27].

The shapes of the fracture surface of the specimens along the braiding direction in the tensile test are shown in Figure 5. After the specimen fractured, the fiber bundles were extracted from the resin matrix. The fiber bundles close to the fracture face were well bonded to the resin. In specimen B-V1 with low fiber volume fraction, for example, most of the surface resins show smooth brittle fractures with many small cracks. In the 3D braided composites, the main function of resin was to transfer load and to withstand interlaminar stresses.

**Table 2.** The main tensile properties of 3D braided composites.

| Specimen | B-V1 | SD/B-V1 | B-V2 | SD/B-V2 | B-V3 | SD/B-V3 |
|---|---|---|---|---|---|---|
| Maximum load/KN | 58.01 | 0.49 | 60.00 | 1.66 | 99.87 | 2.62 |
| Tensile strength/MPa | 580.10 | 1.82 | 600.00 | 2.58 | 745.20 | 1.78 |
| Tensile modulus/GPa | 42.62 | 0.62 | 44.85 | 1.89 | 65.34 | 1.06 |

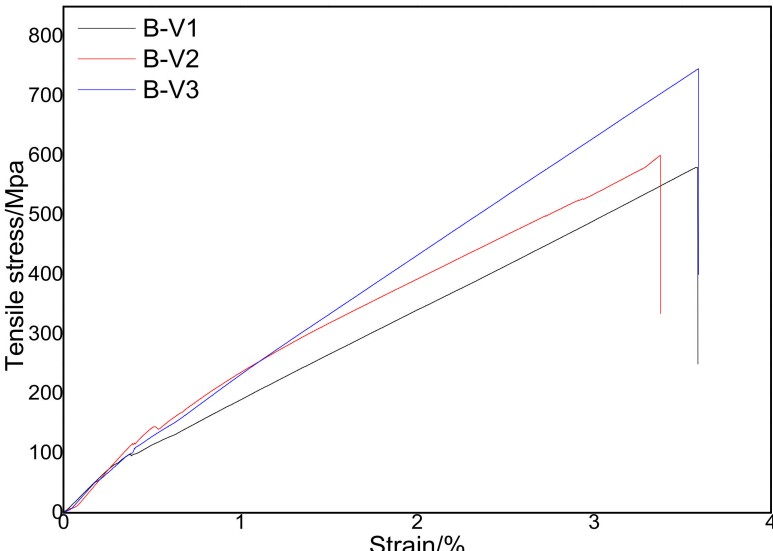

**Figure 4.** Typical tensile stress-strain curves of 3D braided composites.

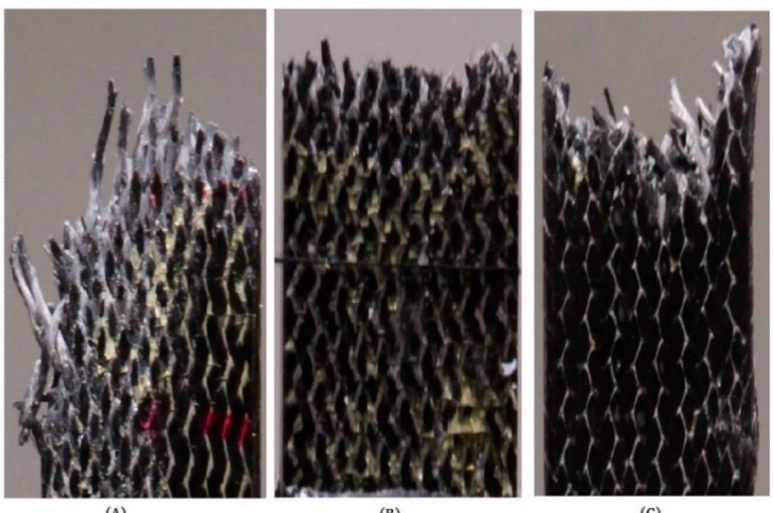

**Figure 5.** Fracture surface shapes of specimens (**A**) B-V1, (**B**) B-V2, and (**C**) B-V3.

### 4.2. AE Signal Analysis of 3D Braided Composites in Tensile Tests

In the tensile testing of the 3D braided composites, the main AE source was the formation and extension of cracks. Because the 3D braided composites were made of fibers and resin, forming an anisotropic multiphase material, when acoustic waves travelled through it, they passed through many modes of refraction, reflection, and transformation of the interfaces. Finally, the acoustic waves arrived at sensors at various times with different speeds. Therefore, the resulting waveforms were complex. In order to study the characteristics of the AE signals at different stages of the stretching process, the AE signal of specimen B-V1 was taken as an example to analyze in detail. The analysis of the load–displacement curves and the AE signal energy parameters of the specimen are shown in Figure 6. As can be seen from Figure 6, the energy distribution of the AE event is obviously agglomeration. In these AE signals, different types of signal had different peak amplitudes, rise times, and durations. This work was based on the statistical toolbox and SOM toolbox in MATLAB R2013a used to implement the clustering analysis of AE data. The boundary values of the cluster analysis results were compared with the damage types corresponding to typical AE signals reported in the literature [28–30]. According to discussion of damage in the literature [31,32], the tensile process can be divided into three stages:

a microcrack initiation (damage initial) stage, a microcrack growth (damage evolution) stage, and an instability and failure (damage) stage.

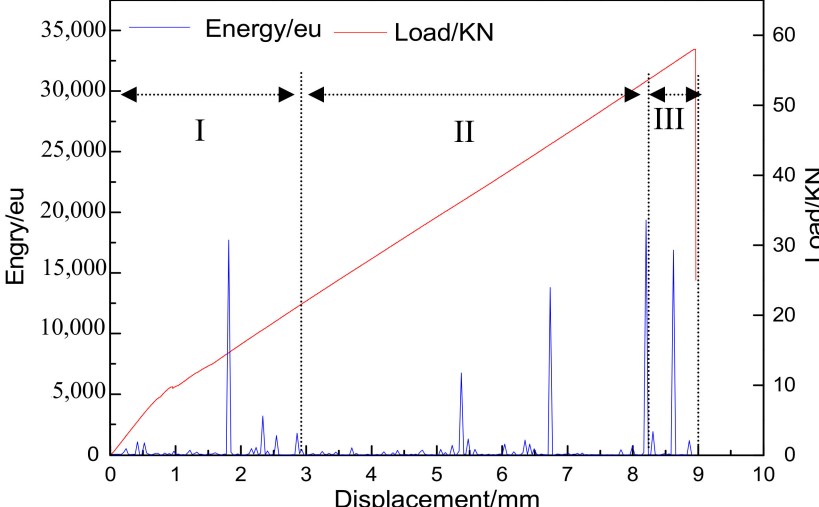

**Figure 6.** Change in the load–displacement curve corresponding with the AE signal energy parameter of specimen B-V1.

During the microcrack initiation (damage initial) stage, with the increase in the load displacement, the AE signal energy parameters continued to accumulate, and the load curve gradually increased. This showed that a small amount of damage to the specimen was generated at the initial loading stage. The main damage features were the specimen's original defects, cracking of the thin surface of the resin matrix, and weak interface debonding. During the microcrack growth (damage evolution) stage, the energy parameters of AE signals fluctuated only slightly, showing that the development of damage was gradual. During the instability and failure (damage) stage, a sudden change occurred in the cumulative energy of AE signal energy. At that point, the failure mode of the specimen was fracture of the fiber bundle accompanied by pull-out of the fiber bundle, leading to the final failure of the specimen.

Taking specimen B-V1 as an example, the corresponding changes in the load–displacement curve and the AE signal ringing count parameter are shown in Figure 7. During the microcrack initiation (damage initial) stage of the specimen, the micro defects in the specimen slowly go through, the microcracks gradually form, and obvious ringing counts begin to appear. At this point, the main AE sources are friction between fibers and matrix, and brittle fracture of the resin on the specimen's surface. With increase in the tensile load, the ringing count also shows an upward trend. During the microcrack growth (damage evolution) stage of the specimen, the ringing count of the specimen is relatively stable, and the peaks appear in only a few positions. This is due to the fiber bundles interwoven into a whole network structure in the interior structure of the 3D braided composites, that make it easier for them to withstand stress concentration at the braided node. When braided yarns are tightened, they will press each other. The interlacing point of braided yarn is called the braided node. The damage occurs essentially at the braided nodes, corresponding to the ringing count peak of the AE signal as shown in Figure 7. Before the crack passes to the next braided node position there is only slow crack propagation. Fiber fracture in the specimen is the main AE source, and the ringing count is less than in the previous stage. Therefore, this stage demonstrates longer sustained load displacement and it is also the most important stage of the tensile process. The cracks in the braided composites increase steadily until the critical size is reached. As the tensile displacement continues to increase, the ringing count rises suddenly. At the failure (damage) stage, specimen B-V1 fractures.

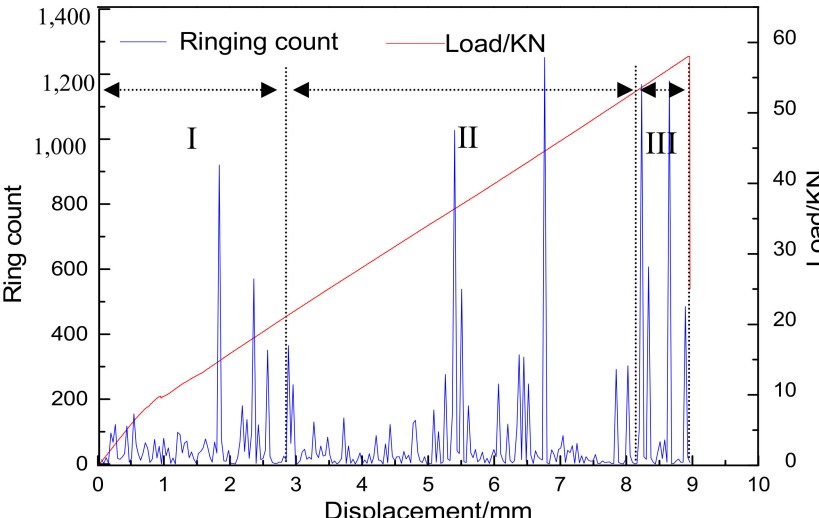

**Figure 7.** Change in the load–displacement curve corresponding with the AE signal ringing count parameter of specimen B-V1.

The load–displacement curve of specimen B-V1 corresponded to the RMS parameter of the AE signal, as shown in Figure 8. During the initial stage of the tensile test, the RMS values of the AE signals fluctuate slightly, showing that the damage is gradually generated in the specimen. The damage mode mainly includes microcracks in the matrix and a gradual increase in cracking at the interface. Fluctuation of the RMS value increases greatly with the increase in tensile load and displacement. At this point, the interface cracks and the crack produced by the matrix becomes large at the braided node. In the interior of the 3D braided composites, the interface between the fiber and resin accelerates cracking, while new crack sources form gradually in other regions. During the microcrack growth (damage evolution) stage, the RMS value is basically stable, evincing peaks at intervals. This shows the process of damage accumulation. At that time the matrix damage is aggravated, local crack propagation occurs, and some fibers begin to break. Finally, the RMS value increases dramatically and the fiber bundle in the specimen fractures.

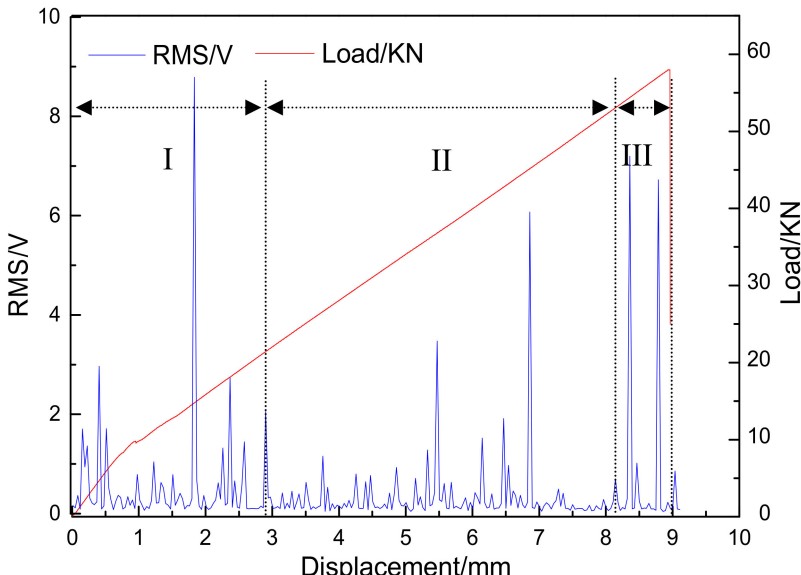

**Figure 8.** Change in the load–displacement curve corresponding with the AE signal RMS parameter of specimen B-V1.

Figure 9 shows the correspondence between the load–displacement curve and the AE signal parameters for B-V2 and B-V3. Compared with B-V1, the development trend of the damage stage for B-V2 and B-V3 is similar. At each stage, only the beginning positions of the damage features differ. Variations in the braiding parameters of 3D braided composites led to the emergence of these differences. With the increase in fiber volume fraction, the displacement at the initial damage stage increases. The higher the fiber volume, the more fiber bundles are responsible for the load.

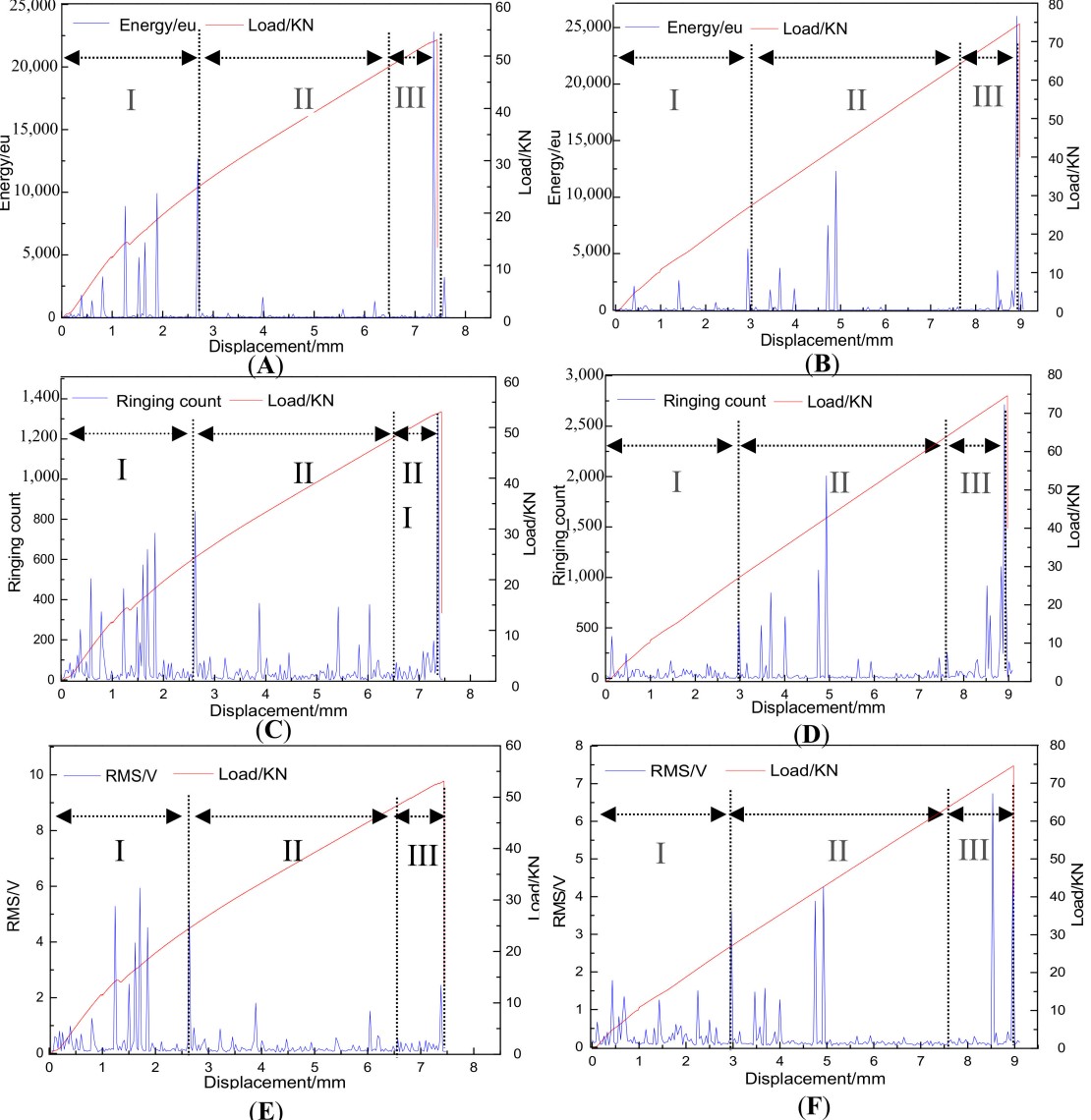

**Figure 9.** Changes in the load–displacement curves corresponding with the AE signal parameters of specimens B-V2 and B-V3. Change in the load–displacement curve corresponding with AE signal energy parameter of (**A**) B-V2 and (**B**) B-V3. Change in the load–displacement curve corresponding with the AE signal ringing count parameters of (**C**) B-V2 and (**D**) B-V3, and changes in the load–displacement curve corresponding with the AE signal RMS parameter of (**E**) B-V2 and (**F**) B-V3.

## 4.3. HHT Analysis of AE Signal in Tensile Test

Deformation of the resin matrix, formation and propagation of cracks, and fracture of fibers could be detected by AE signals. However, by analyzing only the load–displacement curves and AE signal parameters, the characteristic information of the AE signals was difficult to understand intuitively and accurately. It was necessary to analyze the time-frequency locations of those AE signals. Through

analysis, the HHT had strong adaptability to the AE signal. The HHT could reveal the time-frequency characteristics of signals, with good local adaptability and intuitive analysis results.

The original AE signals of the 3D braided composite specimens were decomposed by the EMD of the HHT method to obtain different IMF components from high to low frequencies. The decomposition results are shown in Figure 10, with the abscissa indicating the time and the ordinate indicating the amplitude of the signal. Six IMF sets (imf1-imf6) and one set of the residual component (res.) were separated. In the original signal, different frequency components were separated into imf1~imf6. Among them, imf1 and imf2 were signal noises, and the res. was the residual component. The greater the number of decompositions, the smaller the decomposition oscillations. From the waveform, there was about zero mean line symmetry.

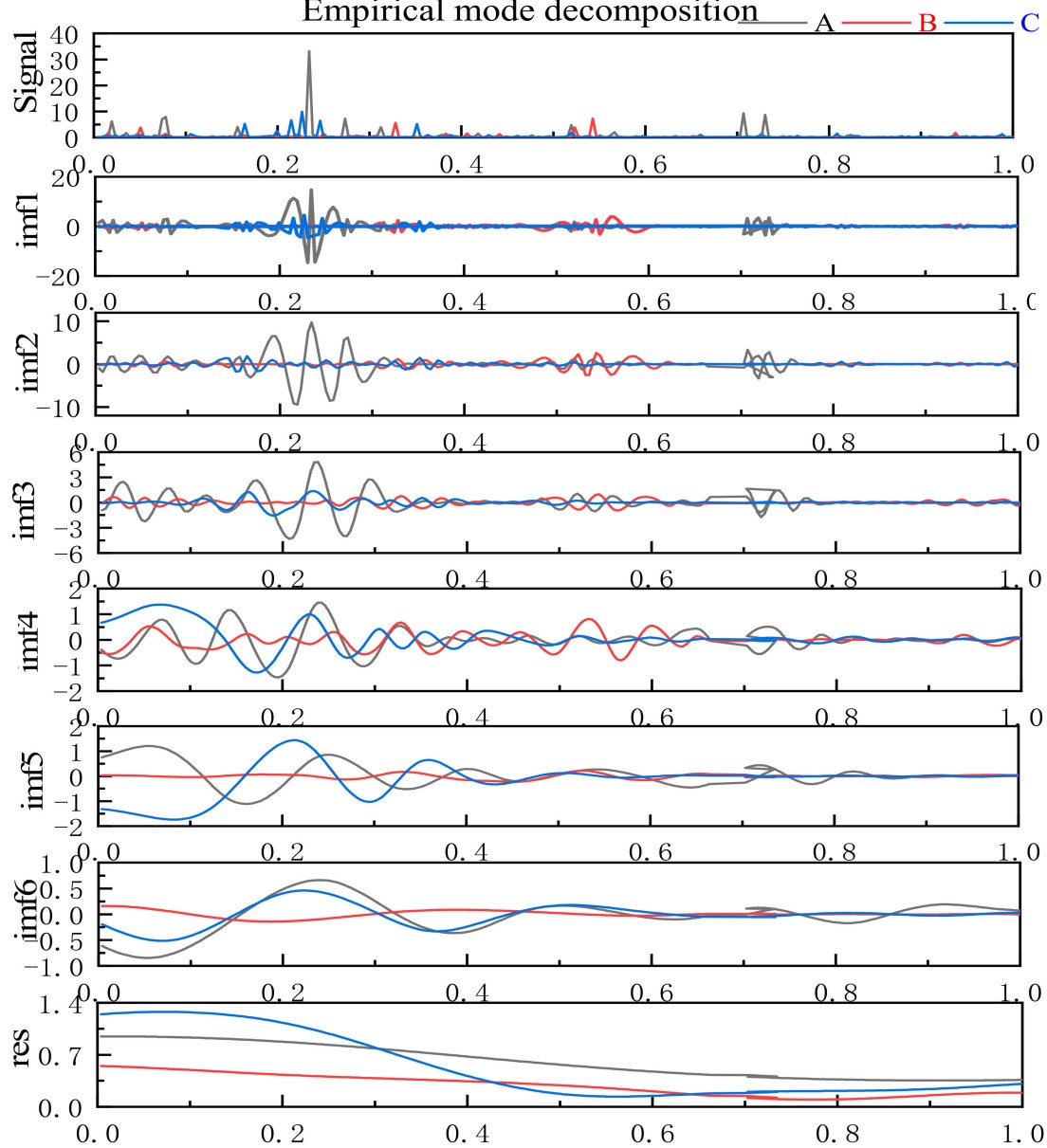

**Figure 10.** Empirical mode decomposition of AE signals for samples (**A**) B-V1, (**B**) B-V2, and (**C**) B-V3.

From Figure 10 it is evident that EMD is an adaptive decomposition based on time scales. Each IMF exhibits a certain range of modes, and there is no mode aliasing between them. At the same time, the IMFs of different scales in the frequency domain are arranged in order from high to low frequencies.

It can be seen that the trends of the EMD diagrams for all specimens are basically the same. Compared with the higher modes, the lower modes of the signal show a significant increase in peak value, indicating that the signal at this stage is mostly an expanding wave.

When the AE signal of the specimen had been decomposed into IMFs, each IMF was transformed by the HHT and the instantaneous frequency of the original signal could be obtained. The HHT spectrum of the AE signal is shown in Figure 11, with the abscissa representing the time of the original signal, the ordinate representing the frequency of the original signal, and the luminance representing the normalized amplitude of the signal, i.e., the magnitude of the signal energy. From the gray level on the right, it can be seen that the brighter the data point, the greater the signal amplitude and the higher the energy. In the tensile process of the 3D braided composites, the energy of AE signal was mainly concentrated below 5 kHz. A total of 6 linear connections are shown in Figure 11, that correspond to the EMD decomposition of the AE signal in Figure 10. Among these, the two linear connections at higher frequencies are blurred, representing the noise part of the original signal. The four linear connections at lower frequencies correspond to the IMFs imf3 ~imf6. With the increase in time, the frequency changes show some vibrations. At the lowest frequency, there is a line-like connection and the frequency change is small. The vibration of AE signal increases with the increase in frequency.

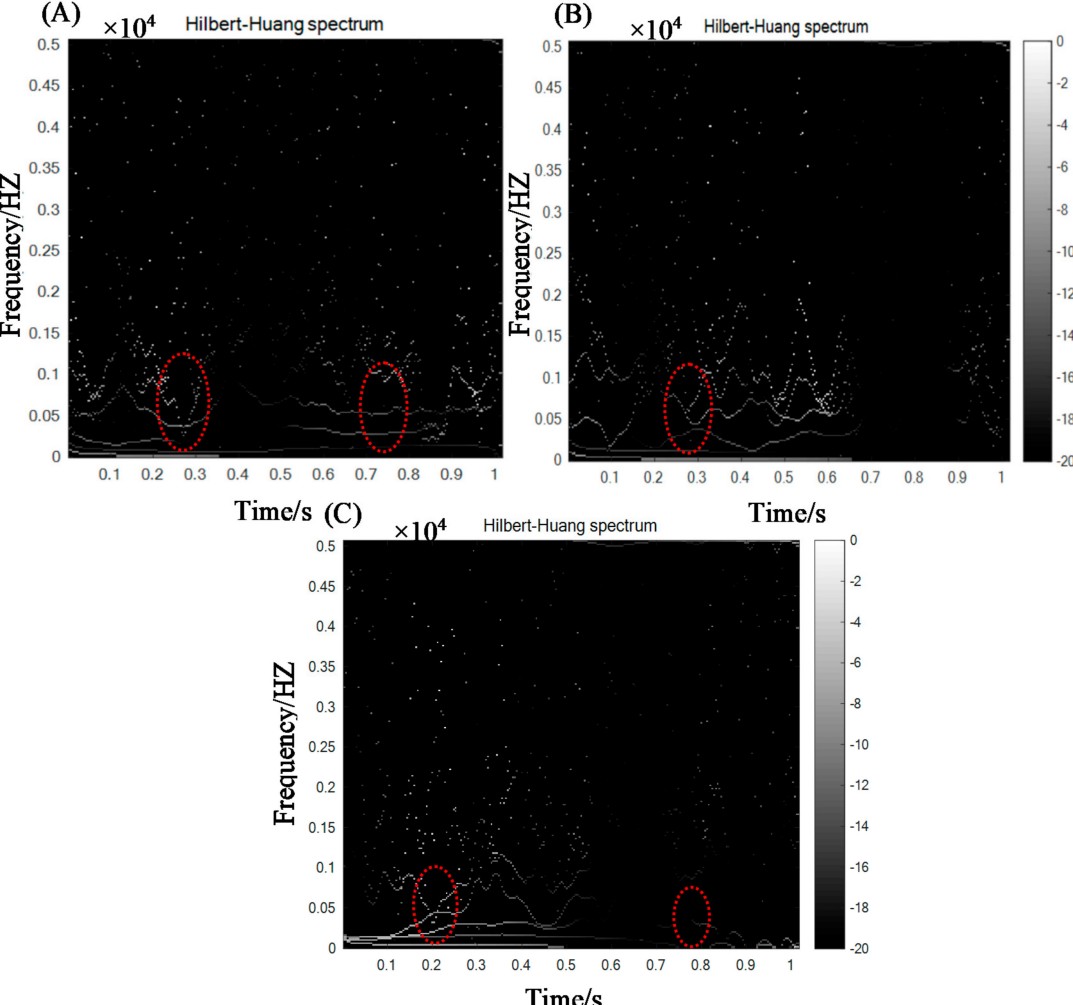

**Figure 11.** Hilbert–Huang transform (HHT) spectra of the AE signal for samples (**A**) B-V1, (**B**) B-V2, and (**C**) B-V3 samples. The areas around the sudden changes in the bright line are circled in red.

From Figure 11, the four peak areas of the frequencies can be observed. The AE signal wave is near to zero in the four corresponding regions. A frequency trough occurs between adjacent frequency peaks

and this is the largest region of energy, that coincides with the peak period of the AE signal. That region corresponds to the microcrack growth (damage evolution) stage. During this stage, the interface between the fiber reinforcement and the matrix begins to undergo debonding. Cracks begin to appear in the resin matrix and the fiber reinforcement also shows some damage. In the HHT spectrum, the time at which the frequency changes from low to high corresponds to the time when the AE signal wave changes during the tensile process. The areas around the sudden change in the bright line that are circled in red indicate the frequency saltation. The frequency saltation implies changes in the failure modes of the 3D braided composites. The corresponding time shows when the failure modes change.

The HHT marginal spectrum of the AE signal is shown in Figure 12 with the abscissa representing the frequency distribution of the stress wave and the ordinate representing the magnitude of the stress wave amplitude. This spectrum clearly reflects that, at the frequencies of 8–12.6 kHz, the signal suddenly changes. The AE signal energy is concentrated in this frequency range, reflecting the good frequency concentration of the HHT marginal spectrum. The amplitude of HHT marginal spectrum increases with the increase in the tensile load. Overall, the amplitudes of the 3D braided composites with different fiber volumes show little difference. The amplitude of the HHT marginal spectrum represents the energy of the signal. From analysis of the failure modes and the HHT marginal spectral amplitudes of the 3D braided composites, it can be concluded that, during the microcrack growth (damage evolution) stage, the amplitude of the HHT marginal spectrum is higher and contains more energy. At that moment, the corresponding damage mode of the 3D braided composites involves cracking of the resin matrix and debonding between resin and the fiber bundle. When the amplitude of the HHT marginal spectrum increases suddenly and the AE increases suddenly, the fiber bundle fractures and the 3D braided composite specimens fail.

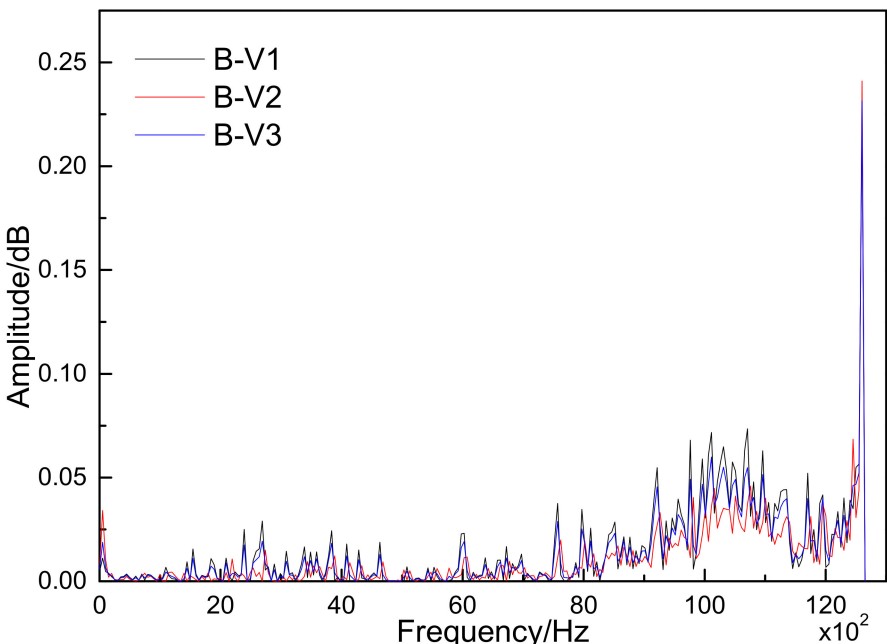

**Figure 12.** HHT marginal spectra of the AE signals for different samples.

## 5. Conclusions

In this study, recognition of damage modes and Hilbert–Huang transform analyses of 3D braided composites with the same braiding angle and different fiber volumes fractions were performed under the tensile condition. To study the time-frequency characteristics of the AE signals, the HHT time-frequency analysis method was used to separate and extract information regarding damage modes. The corresponding relations between the damage modes and AE signals of the 3D

braided composites was obtained. This study provides a new, nondestructive test method for the study and analysis of the tensile fracture mechanisms of 3D braided composites.

**Author Contributions:** Conceptualization, G.D. and J.L.; Methodology, X.P and L.S.; Software, L.S.; Validation, Z.W.; Writing—Original Draft Preparation, G.D. and X.P.; Writing—Review & Editing, Y.T.

**Funding:** This study was funded by the Science & Technology Development Fund of Tianjin Education Commission for Higher Education (Grants No. 2018KJ194).

**Conflicts of Interest:** The authors declare that they have no conflict of interest.

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
