# Peer review of "Recognition of Damage Modes and Hilbert–Huang Transform Analyses of 3D Braided Composites"

_jcs, doi:10.3390/jcs2040065_

Round 1
Reviewer 1 Report
This manuscript presents a comprehensive study on the correlation between acoustic emission signals and damage evolution in braided composites. To this end, a nonlinear and nonstationary signal processing technique, namely Hilbert-Huang transform, is adopted this study and tensile experiments are also performed. The investigated topic is of great importance to the health monitoring of braided composites and fits well within the scope of Journal of Composite Science.
Overall this paper is well written and the referee only has a few minor comments:
(1) Abstract: “the evolution law of 3D braided composites” does not read well. I guess the authors meant the damage evolution law of 3D braided composites.
(2) Line 30: It is better to use “experience” instead of “cause”.
(3) Introduction: The authors mentioned many previous studies on AE technique and pointed out those methods have pros/cons. However, it is not clear what the pros/cons are. It is helpful to be more specific here so readers can better understand the difference between the present and previous studies as well as the innovative contributions of the paper.
(4) Line 59: “Depending to” should be “depending on”.
Author Response
(1) Abstract: “the evolution law of 3D braided composites” does not read well. I guess the authors meant the damage evolution law of 3D braided composites.
Response 1:
Thanks for the reviewer valuable comments.
the damage evolution law of 3D braided composites
(2) Line 30: It is better to use “experience” instead of “cause”.
Response 2:
Thanks for the reviewer valuable comments.
Over a long service period, these composites inevitably experience damage and failure [2-4].
(3) Introduction: The authors mentioned many previous studies on AE technique and pointed out those methods have pros/cons. However, it is not clear what the pros/cons are. It is helpful to be more specific here so readers can better understand the difference between the present and previous studies as well as the innovative contributions of the paper.
Response 3:
Thanks for the reviewer valuable comments.
For a stationary signal, only the time domain characteristics or frequency domain characteristics of the signal are considered to reflect the state of the signal. However, for a typical non-stationary signal of a three-dimensional braided composite acoustic emission signal, it is not only necessary to describe the change of the signal frequency with time, but also a time-frequency joint function is needed to represent the signal, i. e., the time-frequency representation of the signal. As an important method to deal with non-stationary random signals, the time-frequency analysis can effectively identify and remove the interference noise in the signal through the time-frequency spectrum, and comprehensively identify the characteristics of the signal. Typical time-frequency analysis methods include Short-time Fourier Transfom (STFT), Wavelet Transform (WT), S-transform (ST), Gabor expansion, and Hilbert-Huang transform (HHT) and the like.
Compared with the other four time-frequency analysis methods, HHT has a good local adaptability and intuitiveness of analysis results. It has a greater progress in resolution than traditional time-frequency analysis methods, so it is more suitable for three-dimensional braided composite acoustic emission signals. Time-frequency analysis can reveal the time-frequency characteristics unique to acoustic emission signals.
(4) Line 59: “Depending to” should be “depending on”.
Response 4:
Thanks for the reviewer valuable comments.
Depending on the characteristics of the signal

Reviewer 2 Report
The authors present a study on the recognition of damage of 3D braided composites and its transform analysis. The paper looks interesting and well prepared. There are few issues that should be taken into account before consider it for publication. Therefore, the authors are required to revise the manuscript by addressing the following points: - the advantages of the proposed approach with respect to existing ones should be addressed. - english issues can be found locally throughout the manuscript - the authors are required to expand the literature by introducing recent contributions to damage of 3D braided composites: compos struct 2017;163:32-43; compos struct 2017;159:667-676;applied compos mater 2018; 25:823-841 - the quality of fig. 11 should be improved - results shown in fig. 12 are not easy to distinguish between each other. Minor revision is required and the reviewer is willing to review the revision.Author Response
(1) the advantages of the proposed approach with respect to existing ones should be addressed.
Response 1:
Thanks for the reviewer valuable comments.
The same question was raised by the first reviewer. This question has been answered (Response to Reviewer 1 Comments, question 3).
(2) English issues can be found locally throughout the manuscript
Response 2:
Thanks for the reviewer valuable comments.
Checking the manuscript carefully. Refining the language and correcting the grammatical errors and bad words and sentences as possible as we can.
(3) The authors are required to expand the literature by introducing recent contributions to damage of 3D braided composites: compos struct 2017;163:32-43; compos struct 2017;159:667-676;applied compos mater 2018; 25:823-841
Response 3:
Thanks for the reviewer valuable comments. These literatures were carefully read, and which were cited to expand the literature by introducing recent contributions to damage of 3D braided composites.
Zhang et al. [14-16] conducted a meso-scale FEA to predict the mechanical properties and simulate the progressive damage of 3D braided composites under external loadings.
(4) The quality of fig. 11 should be improved
Response 4:
Thanks for the reviewer valuable comments.
The quality of fig. 11 had be improved.
Figure 11. HHT spectra of the AE signal for samples (a) B-V1, (b) B-V2 and (c) B-V3 samples. The areas around the sudden changes in the bright line are circled in red.
(5) Results shown in fig. 12 are not easy to distinguish between each other.
Response 5:
Thanks for the reviewer valuable comments.
The fig. 12 had been revised in order to easy to distinguish between each other.
Figure 12. HHT marginal spectra of the AE signals for different samples.
